# Let GPT be a Math Tutor: Teaching Math Word Problem Solvers with Customized Exercise Generation

**Zhenwen Liang** [✉1], **Wenhao Yu**[2], **Tanmay Rajpurohit**[3],
**Peter Clark**[4], **Xiangliang Zhang**[1], and **Ashwin Kaylan** [✉4]

[1]University of Notre Dame, `zliang6@nd.edu`
[2]Tencent AI Seattle Lab
[3]Georgia Institute of Technology
[4]Allen Institute for AI, `ashwinkv@allenai.org`

## Abstract

In this paper, we present a novel approach for distilling math word problem solving capabilities from large language models (LLMs) into smaller, more efficient student models. Our approach is designed to consider the student model's weaknesses and foster a tailored learning experience by generating targeted exercises aligned with educational science principles, such as knowledge tracing and personalized learning. Concretely, we let GPT-3 be a math tutor and run two steps iteratively: 1) assessing the student model's current learning status on a GPT-generated exercise book, and 2) improving the student model by training it with tailored exercise samples generated by GPT-3. Experimental results reveal that our approach outperforms LLMs (e.g., GPT-3 and PaLM) in accuracy across three distinct benchmarks while employing significantly fewer parameters. Furthermore, we provide a comprehensive analysis of the various components within our methodology to substantiate their efficacy.

## 1 Introduction

Math word problems (MWPs) are an essential aspect of mathematical education and a critical skill to develop for individuals across various domains in life (Hegarty et al., 1995). MWP solving requires the ability to comprehend natural language, extract relevant information, and perform mathematical reasoning to solve the given problem. Research in MWP solvers has gained considerable interest due to the ubiquity of such problems in daily life, ranging from finance and scheduling to engineering and science (Koedinger and Nathan, 2004). Developing AI agents capable of solving MWPs demonstrates an advanced understanding of the interplay between natural language processing and mathematical reasoning, paving the way for practical applications in

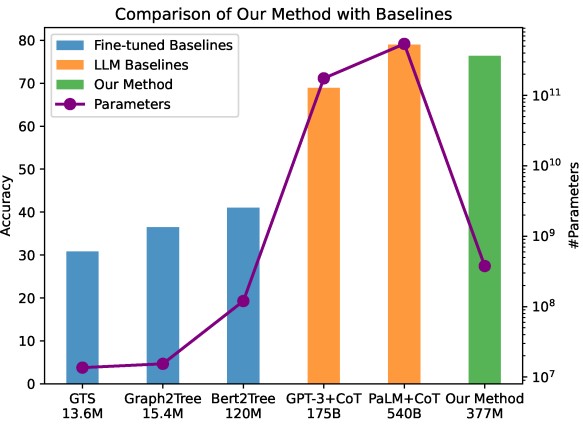

Figure 1: Accuracies vs model sizes for representative baselines and our approach on SVAMP dataset. Our method achieves competitive performance with LLMs with significantly fewer parameters.

education, decision-making, and more (Mukherjee and Garain, 2008). The format of a typical MWP involves a textual description of a problem scenario, which needs to be translated into a mathematical expression (typically an equation) that can be solved to obtain the answer. MWP solving was presented as a task for artificial intelligence several decades ago (Fletcher, 1985). Previous fine-tuned methods usually apply Seq2Seq models (Wang et al., 2017, 2018; Xie and Sun, 2019; Zhang et al., 2020; Liang et al., 2022a). In recent years, large language models (LLMs) such as GPT-3 (Brown et al., 2020) and PaLM (Chowdhery et al., 2022) exhibit strong reasoning ability with the help of chain-of-thought (CoT) prompting (Wei et al., 2022b; Wang et al., 2023; Chen et al., 2022), which achieves striking performance on MWP solving and outperforms fine-tuned state-of-the-art (SOTA) solvers by a large margin.

Despite their remarkable strides in solving MWPs, the substantial number of parameters in LLMs results in computational inefficiency, and necessitates API availability for reproducibility. To

---

This work was done when Zhenwen Liang and Wenhao Yu were interns at the Allen Institute for AI (AI2).

mitigate these issues, a natural solution is distilling the knowledge from LLMs (i.e. **teacher models**) into smaller, more efficient models (i.e. **student models**). The majority of prior research has emphasized using the "explanation" component of the CoT approach as the distilled knowledge (Ho et al., 2022; Li et al., 2022a; Shridhar et al., 2022b; Magister et al., 2022). Nonetheless, these methodologies exhibit certain limitations. Firstly, small student models may struggle to understand CoT explanations, potentially impeding their learning efficacy. Evidence from (Wei et al., 2022b,a) indicates that only exceptionally large language models possess the capability to perform CoT during reasoning. As a result, many student models (Ho et al., 2022; Li et al., 2022a; Magister et al., 2022) trained on explanations do not attain satisfactory accuracy in MWP-solving tasks. Secondly, those distillation processes lack feedback from the student model to the LLM teacher, and thus, neglect to consider the knowledge acquisition state of the student.

In this paper, we also concentrate on harnessing the strengths of LLMs to coach smaller, more efficient MWP solvers, but introduce a novel method for addressing the limitations of previous approaches: **C**ustomized **E**xercise for **MA**th **L**earning (CEMAL). We reframe the problem by shifting our focus from providing explanations for existing exercises (i.e., training set) to *identifying the student model's learning needs* and *generating new exercises tailored to them.* Our CEMAL methodology offers a new perspective on knowledge distillation with LLMs by focusing on customized exercise generation, based on the student models' current learning state. This enables more effective and adaptive learning, addressing limitations in prior approaches that rely heavily on generating more high-quality explanations to existing datasets. This approach offers several advantages: (1) it does not impose CoT ability requirements on small models, allowing them to learn more effectively, (2) it takes into account the learning status of the student model during training.

In fact, our approach CEMAL seamlessly aligns with two fundamental tasks in educational science: *knowledge tracing* and *personalized learning*. Knowledge tracing pertains to monitoring and modeling a student's evolving knowledge state over time (Corbett and Anderson, 1994; Abdelrahman et al., 2023). This process enables the identification of a learner's strengths and weaknesses, usu-

ally by exercises, thereby facilitating the generation of tailored educational experiences. Personalized learning is also of vital importance (Hattie and Timperley, 2007; Grant and Basye, 2014). Being cognizant of the student model's learning status ensures the optimally designed exercises are generated to address the model's specific weaknesses. By monitoring the learning progress, our proposed method can dynamically adapt to the student model's evolving knowledge state, fostering more effective learning outcomes. In our method, we integrate knowledge tracing and learning status into the distillation process to establish a robust connection between the LLM teacher and the student model, yielding a more interactive and customized learning experience. Consequently, this approach substantially enhances the student model's problem-solving capabilities. As illustrated in Figure 1, our knowledge distillation approach achieves competitive accuracy on the SVAMP dataset, but employs significantly fewer parameters compared to state-of-the-art LLMs, such as GPT-3+CoT and PaLM+CoT with few-shot prompting (Wei et al., 2022b).

Our contribution can be summarized as follows:

- We propose a novel method named CEMAL that utilizes LLMs to generate additional data in the form of targeted practice problems, addressing the student model's weak areas.

- Our approach is evaluated on multiple MWP datasets, including both in-distribution (ID) and out-of-distribution (OOD) tests (Koncel-Kedziorski et al., 2016; Miao et al., 2020; Patel et al., 2021). We show that our method is significantly effective in improving student models under the OOD setting.

- The experimental results demonstrate that our method achieves state-of-the-art accuracy, significantly outperforming fine-tuned baselines. Notably, the student model trained with our method even surpasses LLMs with few-shot CoT prompting, despite having significantly fewer parameters.

## 2 Related Work

### 2.1 Math Word Problem Solving

After many years of research on rule-based algorithms (Hosseini et al., 2014; Mitra and Baral, 2016) and semantic parsing methods (Shi et al.,

2015; Huang et al., 2017), deep learning has become the predominant technique for solving MWPs, thanks to its superior performance and better generalization ability. Deep neural solver (DNS) (Wang et al., 2017) was among the first to apply Seq2Seq models with RNN encoder and decoder for MWP solving, and subsequent research has mainly explored different structures of RNN-to-RNN solvers (Wang et al., 2018; Liu et al., 2019a; Xie and Sun, 2019; Li et al., 2019; Zhang et al., 2020; Liu et al., 2021). More recently, pre-trained language models such as BERT (Devlin et al., 2019) and RoBERTa (Liu et al., 2019b) have demonstrated remarkable language understanding abilities, leading researchers to replace MWP encoders with these models (Li et al., 2022b; Huang et al., 2021; Shen et al., 2021; Patel et al., 2021; Liang et al., 2022a), resulting in significant accuracy improvements. Recently, LLMs have shown great success in MWP solving, with much superior accuracy compared to fine-tuned baselines, simply by being provided with a few CoT examples of the problem-solving processes. Interestingly, researchers also found that LLMs can reason with zero-shot prompts such as "Let's think step by step" (Kojima et al., 2022). Currently, numerous studies have been conducted to improve the performance of LLMs, including recent works by (Wang et al., 2023; Chen et al., 2022; Zhou et al., 2023; Diao et al., 2023; Liang et al., 2023).

In this paper, we leverage the great success of LLMs to facilitate the training of the student solver. Our approach utilizes an efficient fine-tuned solver as its backbone and yields even superior performance to LLMs, further pushing the limit of deep learning methods in MWP solving.

## 2.2 Large Language Models for Knowledge Distillation and Data Generation

In recent years, there has been a surge of interest in knowledge distillation from LLMs to smaller models. Due to the unavailability of model structure for LLMs, their application is often limited to prompt design and subsequent data generation. Therefore, data generation has become an integral part of knowledge distillation in the context of LLMs. Several studies have investigated the potential of LLMs in knowledge distillation and data generation. For instance, PromDA (Wang et al., 2022) applies prompt-based data augmentation to low-resource natural language understanding tasks, and

AugESC (Zheng et al., 2022) leverages the GPT-J (Wang and Komatsuzaki, 2021) model and utilizes publicly available dialog posts to trigger conversations on various topics. Then, West et al. (2022) employs a generalized LLM to create common sense knowledge graphs, while WANLI (Liu et al., 2022) combines LLM-generated examples with human evaluation to establish diverse natural language inference datasets. Additionally, ZeroGen (Ye et al., 2022b) proposes a zero-shot learning approach by generating datasets using pre-trained LLMs, Pro-Gen (Ye et al., 2022a) uses the quality of generated samples as feedback to LLMs to improve the data generation, and Shao et al. (2023) feeds chain-of-thought demonstrations to LLMs and targets generating more exemplars for in-context learning.

In the domain of MWP solving, several studies have also been conducted with the objective of distilling the reasoning capability of LLMs into smaller solvers by employing chain-of-thought explanations. (Ho et al., 2022) introduces Fine-tune-CoT, which uses LLMs to generate reasoning step instances, subsequently facilitating the fine-tuning of smaller models. (Li et al., 2022a) explores three explanation generation strategies and incorporates them into a multi-task learning framework tailored for compact models. (Magister et al., 2022) assesses the efficacy of chain-of-thought explanations in the training of a small model across three disparate tasks, namely arithmetic reasoning, commonsense reasoning, and symbolic reasoning. Furthermore, (Shridhar et al., 2022b) presents Decompositional Distillation, an approach that segments the problem into subproblems to enhance smaller models' performance.

In contrast to these existing works, our proposed knowledge distillation approach in MWP solving is unique in that it does not focus on the chain-of-thought explanation and it takes into account the learning status of the student model and generates exercises that tailor to the specific weaknesses of the student. Our approach bridges the gap between knowledge distillation and data augmentation in the context of math word problem solvers, allowing student models to improve their problem-solving capabilities more effectively.

## 3 Approach

### 3.1 Problem Definition

Our objective is to train a student Math Word Problem (MWP) solver with the assistance of large lan-

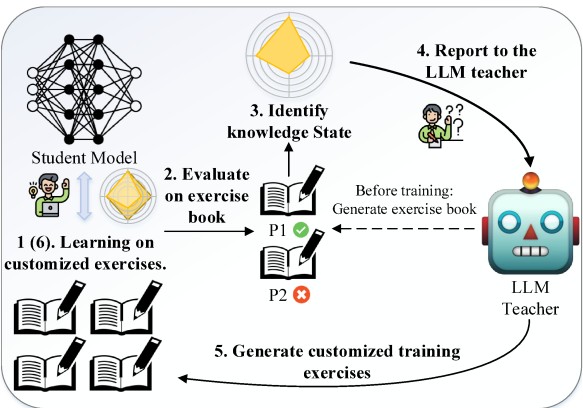

Figure 2: This figure shows the overall iterative framework of CEMAL. After one round of training, the student, which is a small MWP solver, is evaluated by exercises provided by an LLM teacher. Subsequently, LLM generates customized exercises that target the student's knowledge state and weaknesses, thereby facilitating a customized improvement in their overall performance.

guage models (LLMs). The student MWP solver takes a textual description $W$ as input and produces an equation $A$ as output, which indicates the solution process to arrive at the final answer. We represent quantities in $W$ and $A$ using placeholders following number mapping (Wang et al., 2017), which unifies the representation of quantities in different MWPs. Specifically, $W$ and $A$ do not contain the actual values of quantities, but are denoted as $N_1, N_2, ..., N_k$, where $N_i$ refers to the $i$-th number, and $k$ is the maximum number of quantities in $W$ and $A$. This approach offers two advantages: (1) it is a widely used data preprocessing technique in MWP solving that unifies the representation of quantities and reduces the vocabulary size, and (2) during exercise generation, where the goal is to generate MWP variants, number mapping prevents variants generated solely by changing the values of quantities.

### 3.2 Training Workflow

Our proposed training workflow is an iterative and progressive framework that integrates LLMs into the training procedure of student MWP solvers as shown in Figure 2. Before training, we augment the training set $\mathcal{T}_{train}$ by generating $m$ more problems than the original training set, to enlarge the training set at the beginning to push the limit of the student solvers (lines 1-3 in Alg. 1). We also generate $n$ times more problems to formulate the exercise book $\mathcal{X}$ (lines 4-6 in Alg. 1). During training, we perform model validation on $\mathcal{X}$ and select the

---

**Algorithm 1** Training Framework

**Input**: Training Set $\mathcal{T}_{train}$, Exercise Generation Function $EG$, Exercise Book $\mathcal{X}$, Student Solver $S$
**Parameters**: Probability Threshold $\lambda$, number of generations $m, n, k$

1: **for** Each $W$-$A$ pair in $\mathcal{T}_{train}$ **do**     ▷ Before Training
2:     $\mathcal{T}_{train} \leftarrow \mathcal{T}_{train} + EG(W, A, m)$     ▷ Initial Augmentation by $m$ times
3: **end for**
4: **for** Each $W$-$A$ pair in $\mathcal{T}_{train}$ **do**
5:     $\mathcal{X} \leftarrow EG(W, A, n)$     ▷ Generate an exercise book with $n$ times the size of the training set
6: **end for**
7: **while** Training **do**                    ▷ Training starts
8:     **for** Each $W$-$A$ pair in $\mathcal{T}_{train}$ **do**
9:         Train Solver $S$ with $W$-$A$
10:     **end for**
11:     **for** Each $W$-$A$ pair in $\mathcal{X}$ **do**
12:         **if** $S$ cannot solve $W$ **then**
13:             $p \leftarrow U \sim \text{Uniform}(0, 1)$
14:             **if** $p \leq \lambda$ **then**
15:                 $\mathcal{T}_{train} \leftarrow \mathcal{T}_{train} + EG(W, A, k)$
16:             **else**
17:                 Randomly sample a $\tilde{W}$ - $\tilde{A}$ pair from $\mathcal{X}$
18:                 $\mathcal{T}_{train} \leftarrow \mathcal{T}_{train} + EG(\tilde{W}, \tilde{A}, k)$
19:             **end if**
20:         **end if**
21:     **end for**
22: **end while**

---

problems that the solver is unable to solve, which reflect the knowledge state of the student at that time. Subsequently, we generate customized exercises and incorporate them into the original training set $\mathcal{T}_{train}$ to update it. Specifically, we generate $k$ exercises for every source MWP (the details of exercise generation are presented in section 3.4). This iterative process allows for the continuous expansion of the training set and leads to enhanced performance of the student solver. Furthermore, we introduce a threshold $\lambda$ to control the proportion of targeted generation compared to random generation. A detailed description of $\lambda$ can be found in Section 3.5. The pseudo-code of our approach is located in Alg. 1. This framework closely mimics the knowledge tracing method in human learning and thus holds promise for enhancing the effectiveness of future educational practices.

### 3.3 Backbone of Student Solver

In this paper, we employ a Seq2Seq model with the Goal-driven Tree-based Solver (GTS) (Xie and Sun, 2019) as our decoder, which has been widely applied in MWP solving and shown to outperform Transformer decoders (Lan et al., 2022). For the encoder part, we use three different backbones - LSTM (Hochreiter and Schmidhuber, 1997),

RoBERTa-base, and RoBERTa-large (Liu et al., 2019b) to demonstrate the backbone-agnostic nature of our approach.

## 3.4 Exercise Generation

Our method generates targeted exercises and their answers based on a given source problem-answer pair ($W$ and $A$), which is similar to data augmentation. Following a previous paper on analogical reasoning (Liang et al., 2022b), the generation of exercise targets to reach two kinds of analogies to $W$ and $A$ - problem analogy and solution analogy - that help models better understand MWPs.

For problem analogy, we generate problems with similar problem descriptions but different solutions. Our method perturbs not only questions but also considers the context, and we prompt LLMs with few-shot examples to generate MWP variants. In contrast to previous studies that used human annotators (Patel et al., 2021; Yang et al., 2022), our approach is automated and scalable. For solution analogy, we generate problems with different keywords in the problem description but a similar solution to the source MWP. Interestingly, we find that LLMs can achieve this generation in a zero-shot manner by simply prompting them to generate problems similar to the source problem. Our prompt design is located in the appendix.

This exercise generation is deployed at two different steps in our proposed training framework. First, we generate an exercise book by augmenting the entire training set (lines 4-6 in Alg. 1), which serves as a validation set to identify the weaknesses of the student solver. We do not directly use the training set to validate the student solver because we want a more diverse validation set to comprehensively evaluate the student model. Additionally, the solver may memorize the MWPs in the training set instead of fully understanding them, as previous research has found that slightly perturbed training MWPs can cause a solver to fail (Liang and Zhang, 2021). Therefore, validation on the exercise book, which contains many variants of the training set, provides a more robust evaluation.

Secondly, after identifying the problems that the student solver cannot solve from the exercise book, we use them as the source to generate customized exercises and add them to the previous training set (lines 12-16 in Alg.1). In this way, the training set grows progressively and covers more knowledge, leading to a stronger student MWP solver.

The exercise problems generated by LLMs may be of low quality, in a wrong format, or repetitive (a case study including correct and incorrect examples is presented in Section 4.7). To alleviate the impact of these low-quality exercise problems, we filter out about 30% of problems with the wrong formats during the generation process by checking whether the generated problem is repetitive and the generated answer is a valid mathematical equation.

## 3.5 Targeted vs. Random Generation

The generation of targeted exercises to address MWPs that the student model has previously failed to solve significantly improves the model's capacity to process problem and answer analogies. To evaluate the efficacy of this approach, it is insightful to consider a baseline generation strategy, namely random generation (referenced in lines 17-18 of Alg. 1). We introduce a probability threshold $\lambda$ to control the employment of generation strategy when augmenting the training set (lines 13-18 in Alg. 1). $\lambda = 0$ favors random generation, whereas $\lambda = 1$ mandates the utilization of targeted generation. A detailed analysis on the impact of $\lambda$ can be found in Section 4.4.

## 4 Experiments

### 4.1 Datasets

**MAWPS** The MAWPS dataset (Koncel-Kedziorski et al., 2016) is an aggregation of 2,373 English Math Word Problems (MWPs) from various sources, including AddSub (Hosseini et al., 2014), SingleOp (Roy et al., 2015), MultiArith (Roy and Roth, 2015), SingleEq (Koncel-Kedziorski et al., 2015), and SimulEq (Kushman et al., 2014). We employ a 5-fold cross-validation for the evaluation on this dataset.

**ASDiv-a** ASDiv (Miao et al., 2020) is an English MWP dataset designed to exhibit a more diverse range of language patterns and problem types, comprising 2,305 MWPs. In accordance with prior studies (Patel et al., 2021; Lan et al., 2022), we select the arithmetic subset, ASDiv-a, which contains 1,218 MWPs and utilizes a 5-fold cross-validation method for evaluation.

**SVAMP** The SVAMP dataset is a test-only dataset (Patel et al., 2021), consisting of 1,000 English MWPs generated by introducing challenging variations to existing problems. We adopt the two evaluation settings proposed in (Patel et al.,

| | MAWPS (ID) | ASDiv-a (ID) | SVAMP (ID) | SVAMP (OOD) |
|---|---|---|---|---|
| Prior best (Fine-tuning) | $92.0/121M^a$ | $82.2/144M^b$ | $65.0/144M^b$ | $47.3/121M^a$ |
| Prior best (Knowledge Distillation) | $94.5/11.3B^c$ | – | – | $20.7/6.7B^d$ |
| Few-shot CoT (Wei et al., 2022b) | $93.3/540B$ | $93.1/175B^*$ | $79.0/540B$ | $\mathbf{79.0}/540B$ |
| Without CEMAL — LSTM (20M) | 82.6 | 71.4 | 45.0 | 30.8 |
| Without CEMAL — Base (144M) | 88.5 | 81.2 | 69.2 | 41.0 |
| Without CEMAL — Large (377M) | 90.4 | 87.6 | 78.5 | 49.5 |
| CEMAL (Our Solvers) — LSTM (20M) | 92.0 | 86.9 | 67.1 | 53.4 |
| CEMAL (Our Solvers) — Base (144M) | 93.9 | 90.9 | $81.5$ | 68.6 |
| CEMAL (Our Solvers) — Large (377M) | **94.7** | **93.3** | **85.4** | $76.4$ |

Table 1: We compare the accuracy and number of parameters on 4 benchmarks in the format of (accuracy/number of parameters). Prior best baselines are the following. a: (Jie et al., 2022), b: (Patel et al., 2021), c: (Magister et al., 2022), d: (Ho et al., 2022). On each dataset, the best performance is **bolded** and the second best is in *italics*. [*]: This accuracy is calculated on the ASDiv-a subset out of ASDiv based on the results in `https://github.com/jasonwei20/chain-of-thought-prompting`. ID denotes the in-distribution test and OOD denotes the out-of-distribution test.

2021). The first setting employs a 5-fold cross-validation approach on the 1,000 MWPs, incorporating MAWPS (Koncel-Kedziorski et al., 2016) and ASDiv-a (Miao et al., 2020) as additional training data for each fold. In the second setting, MAWPS and ASDiv-a serve as the training set, while SVAMP is used as the testing set.

**Evaluation Setting** We categorize the evaluations on MAWPS, ASDiv-a, and the first setting on SVAMP as in-distribution (ID) tests, as they all involve 5-fold cross-validation on a specific dataset. Conversely, the second setting on SVAMP is considered an out-of-distribution (OOD) test, given that the training and testing sets originate from distinct sources.

### 4.2 Implementation

The backbone of our pre-trained encoder is a RoBERTa model (Liu et al., 2019b). For LLM, we use the ChatGPT *gpt-3.5-turbo* API to perform problem generation. To encourage a more diverse generation, we set the temperature to 1.25. All the experiments in this paper can be conducted with a cost lower than 100 dollars on OpenAI API calls. For evaluation, we use the accuracy of answer value as our evaluation metric following all the baselines. Since the answer value can be calculated from different equations, it is more reasonable to check if the value is correct, rather than checking the element-wise equivalence of a generated equation and the corresponding ground truth equation.

We conducted all experiments using an NVIDIA

RTX A6000 graphics card, implemented in Python using the PyTorch framework. The training was performed for 50 epochs with a batch size of 16, using the AdamW (Kingma and Ba, 2015; Loshchilov and Hutter, 2018) optimizer with an initial learning rate of 8e-6 for the pre-trained model part (i.e. RoBERTa) and 5e-4 for the GTS decoder, which was halved every 20 epochs. The weight decay during training was set to 0.01, and a dropout rate of 0.25 was applied to the decoder to prevent overfitting. The training set will be augmented during epoch 10 and epoch 20, i.e., lines 8-18 in Alg. 1 will only be executed at epoch 10 and 20. $m$ and $n$ in Section 3.2 are set to 20 and 4, respectively, with the aim of pushing the limit of small math word problem solver accuracies and achieving higher performance. This is orthogonal to the contribution of this paper because our proposed customized generation strategy and exercise book are demonstrated to be effective in Sections 4.4 and 4.5. Although we understand that further increasing the occurrences of validation and augmentation will further boost our accuracy, we limit their magnitude for better efficiency and lower cost on API calls. During validation, we generate 2 zero-shot and 2 few-shot similar problems for each source problem in the exercise book, therefore $k = 4$ in Alg. 1.

### 4.3 Comparison with Baselines

In order to demonstrate the effectiveness of our knowledge distillation approach, we compare our method with several strong baselines, including the

| Dataset | Backbone | In-Distribution | | |
|---|---|---|---|---|
| | | Random | Half | Target |
| ASDiv-a | GTS | 77.5 | 77.8 | **79.0** |
| | RoBERTa-base | 84.3 | **85.2** | 84.6 |
| | RoBERTa-large | 90.1 | 90.3 | **90.6** |
| MAWPS | GTS | 89.2 | 88.9 | **89.3** |
| | RoBERTa-base | 90.3 | 90.3 | **91.0** |
| | RoBERTa-large | 91.4 | 92.3 | **92.8** |
| SVAMP | GTS | 47.7 | 49.6 | **50.3** |
| | RoBERTa-base | 72.1 | 72.8 | **73.3** |
| | RoBERTa-large | 80.8 | 79.9 | **80.9** |
| Dataset | Backbone | Out-of-Distribution | | |
| | | Random | Half | Target |
| SVAMP | GTS | 33.6 | 36.3 | **38.2** |
| | RoBERTa-base | 50.1 | 50.8 | **55.3** |
| | RoBERTa-large | 62.9 | 63.1 | **65.0** |

Table 2: Results of different problem generation strategies on three datasets under in-distribution and out-of-distribution (OOD) testing. Boldface indicates the best result among the three strategies.

best fine-tuned baseline, the best LLM-enhanced knowledge distillation baseline, and few-shot chain-of-thought (CoT) prompting (Wei et al., 2022b), using the MAWPS, ASDiv-a, and SVAMP datasets. The results presented in Table 1 show that our approach outperforms all the baselines on the MAWPS and ASDiv-a datasets, achieving 94.7% and 93.3% solving accuracy, respectively. For the SVAMP dataset, our approach outperforms the best LLM-enhanced knowledge distillation baseline, achieving 85.4% accuracy on the SVAMP (ID) dataset, which is a significant improvement over the prior best accuracy of 65.0% achieved by fine-tuning. On the SVAMP (OOD) dataset, our approach achieves a solving accuracy of 76.4%, which is lower than CoT-based LLMs, but much higher than the fine-tuned baselines. We also show the original performance of backbone solvers without any additional exercises. To obtain best performance in Table 1, our solvers use about 20x more MWPs than the original training set to train. Overall, our results indicate that our knowledge distillation approach achieves superior performance with much fewer parameters than the prior state-of-the-art LLMs, and outperforms the best fine-tuning and knowledge distillation baselines on all datasets.

| | Training set as exercise book | Same size as training set | n = 2 | n = 4 |
|---|---|---|---|---|
| Base | 54.6 | 59.8 | 60.4 | 65.2 |
| Large | 64.7 | 69.8 | 70.2 | 72.7 |

Table 3: Performance comparison of using training set and different sizes of exercise book for validation on SVAMP dataset.

## 4.4 Analysis on Generation Strategy

The performance of student solvers can be significantly impacted by the generation strategies employed to create the exercise problems. In this analysis, we explore the effectiveness of three different generation strategies: random, half, and target (using threshold values of $\lambda = 0, 0.5, 1$, respectively) on three datasets, in both in-distribution and out-of-distribution settings. Our goal is to identify the best strategy for maximizing the student model's performance. We remove the initial augmentation (setting $m = 0$) and lower the number of generations in this analysis to improve the efficiency of the experiments. Therefore, the results presented in Table 2 differ from our best results in Table 1. Our analysis reveals that the targeted generation strategy mostly outperforms the other two strategies. This suggests that our proposed targeted generation is an effective approach for identifying the weak areas of student solvers and improving them in a customized way.

Moreover, we can clearly see that the improvement is more noticeable in the OOD scenario. A possible reason for this could be that in the ID situation, where the training and testing sets have some shared knowledge components, using random generation for the source problems in the training set also helps to enhance the performance on the testing set. On the other side, in the OOD scenario, where there's a large gap between the training and testing sets, our approach of creating tailored exercises specifically targets the weak points of the student model, leading to a more effective boost in its accuracy.

## 4.5 Analysis on Exercise Book

We conducted an experiment to demonstrate the effectiveness of our proposed exercise book by replacing the original training set with the exercise book and changing the size $m$. As illustrated in Table 3, the performance of generating a same-size exercise book is significantly better than that of us-

| Case 1: Equivalent problem generation | |
|---|---|
| This case shows that LLM generates an equivalent problem using different keywords. | |
| **Source Problem** | In a school there are N0 girls and N1 boys. How many more girls than boys does the school have? |
| **Answer** | N0 - N1 |
| **Generated Problem** | In a zoo there are N0 lions and N1 tigers. How many more lions than tigers are there in the zoo? |
| **Answer** | N0 - N1 |
| Case 2: Sub-problem generation | |
| This case demonstrates an instance where a sub-problem is derived from the source problem. | |
| **Source Problem** | In a school there are N0 girls and N1 boys. N2 more girls joined the school. How many more girls than boys does the school have now? |
| **Answer** | N0 + N2 - N1 |
| **Generated Problem** | In a school there are N0 girls and N1 boys. N2 more girls joined the school. How many girls are there in total now? |
| **Answer** | N0 + N2 |
| Case 3: Incorrect format generation | |
| This case shows a problem where the LLM generates a problem with incorrect formatting. | |
| **Source Problem** | For Gwen's birthday she received N0 dollars. She spent some money and has N1 dollars left. How much money did she spend? |
| **Answer** | N0 - N1 |
| **Generated Problem** | For Gwen's birthday she received N0 dollars. She spent some money and now has N1 dollars left. How much money did she originally have? |
| **Answer** | N0 = N1 + spent amount (Wrong format) |

Table 4: Examples of problem generation by our method. Case 1 and Case 2 (colored in green) show successful instances where mathematically equivalent problem and sub-problem are generated respectively. Case 3 (colored in red) illustrates a problem with incorrect formatting.

ing the training set itself for validation. Validating on the training set cannot fully reflect the learning status of the student model, which may simply memorize the solutions of training problems and is not robust to slightly perturbed problems. This strongly confirms that our proposed exercise book approach can identify weaknesses of the student model. Furthermore, by increasing the exercise book size, more knowledge can be covered, leading to further improvement in accuracy. It is worth noting that the "Same Size as Training Set" method has a similar performance to "n = 2" due to the removal of problems with wrong formats and duplicated problems, as stated in Section 3.4. Therefore, the size of the exercise book under these two settings is empirically similar.

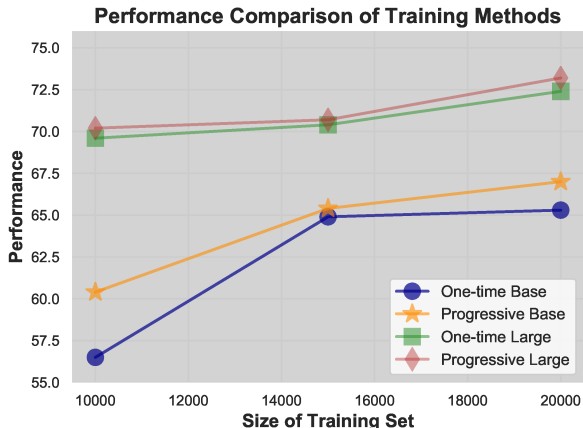

Figure 3: Performance Comparison between one-time augmentation and progressive augmentation on SVAMP under out-of-distribution setting.

### 4.6 Analysis on Progressive Augmentation

To evaluate the efficacy of our proposed progressive and customized training framework, we compare it against the one-time data augmentation approach. Specifically, one-time data augmentation means that, we augment the size of the training set at the beginning of the training process to be the same as the final size of the training set in our proposed framework and evaluate the performance of the student MWP solver on SVAMP-OOD. Intuitively, the one-time augmentation will be better than progressive augmentation because it accesses the entirely augmented training set earlier. However, as shown in Figure 3, the results indicate that our progressive training method outperforms the one-time data augmentation approach in terms of enhancing the performance of the student solver.

## 4.7 Case Study

In Table 4, we present several example problems generated by our method, comprising both high-quality and problematic outputs. The first case demonstrates the LLM's ability to produce a mathematically equivalent problem using distinct keywords. The second successful instance yields a sub-problem derived from the source problem, which has been proven to facilitate a better understanding of mathematical word problems for models (Shridhar et al., 2022a). However, not all generated problems are ideal. As illustrated in the third example, our method occasionally generates MWPs with incorrect formatting, rendering them unsuitable for training our solver. As a result, we filter out such outputs to maintain the quality and accuracy of our training data, by checking if the generated answer is a valid mathematical equation.

## 5 Conclusion

In this work, we present a novel approach CEMAL to use large language models to facilitate knowledge distillation in math word problem solving. Our method first generates an exercise book to evaluate student models and then provides additional training exercises that are customized to the learning needs of the student model, thereby improving the student solver's ability to solve MWP. Our extensive experimental results demonstrate that our proposed approach outperforms all fine-tuned and knowledge distillation baselines on all datasets, while achieving competitive performance against LLMs with significantly fewer parameters. Additionally, we explore different selection generation strategies, revealing that our proposed customized generation strategy is the most effective method, especially in the in-distribution setting. In our future work, we aim to extend this approach to other NLP tasks to evaluate its generalization capability.

## Limitations

Firstly, our approach necessitates meticulous prompt design to generate exercises, which inevitably entails human intervention. This aspect could introduce potential bias or variability and may not scale efficiently.

Secondly, we have not explicitly addressed the quality and correctness of the generated problems.

Our current filtering process only eliminates problems with incorrect formatting. Thus, there exists a significant opportunity for enhancing the effectiveness of our approach by incorporating mechanisms for evaluating and ensuring the quality and correctness of the generated exercises.

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

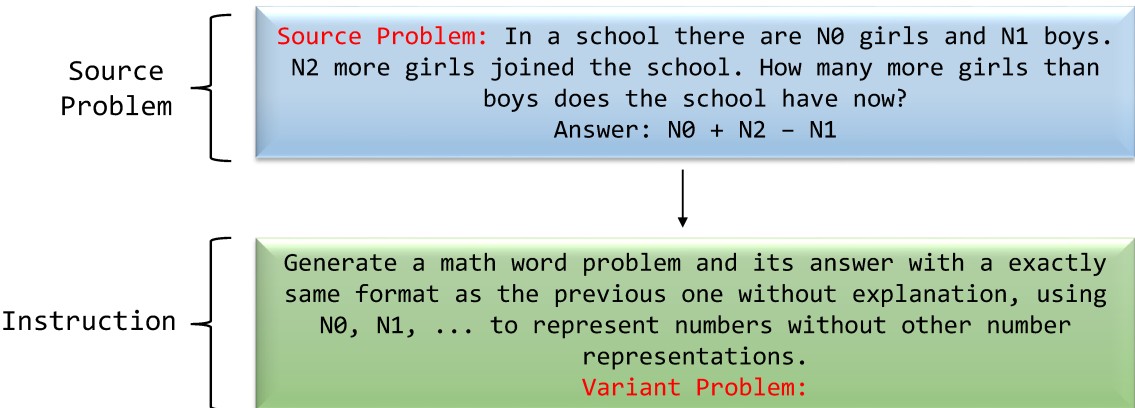

Figure 4: The zero-shot prompts to generate solution-analogy MWP pairs.

## A   Appendix

### A.1   Prompts

We have two different prompting strategies, i.e., few-shot prompting and zero-shot prompting. The former gives the LLM examples of how to generate a variant problem given a source problem, as shown in Figure 5. The full list of exemplars can be found in Table 5. The latter only use instructions to guide LLM in generating a similar MWP, as shown in Figure 4. The examples of generated problems can be found in Table 4. Empirically, we use the ChatGPT *gpt-3.5-turbo* API with a temperature of 0.9 to perform generation.

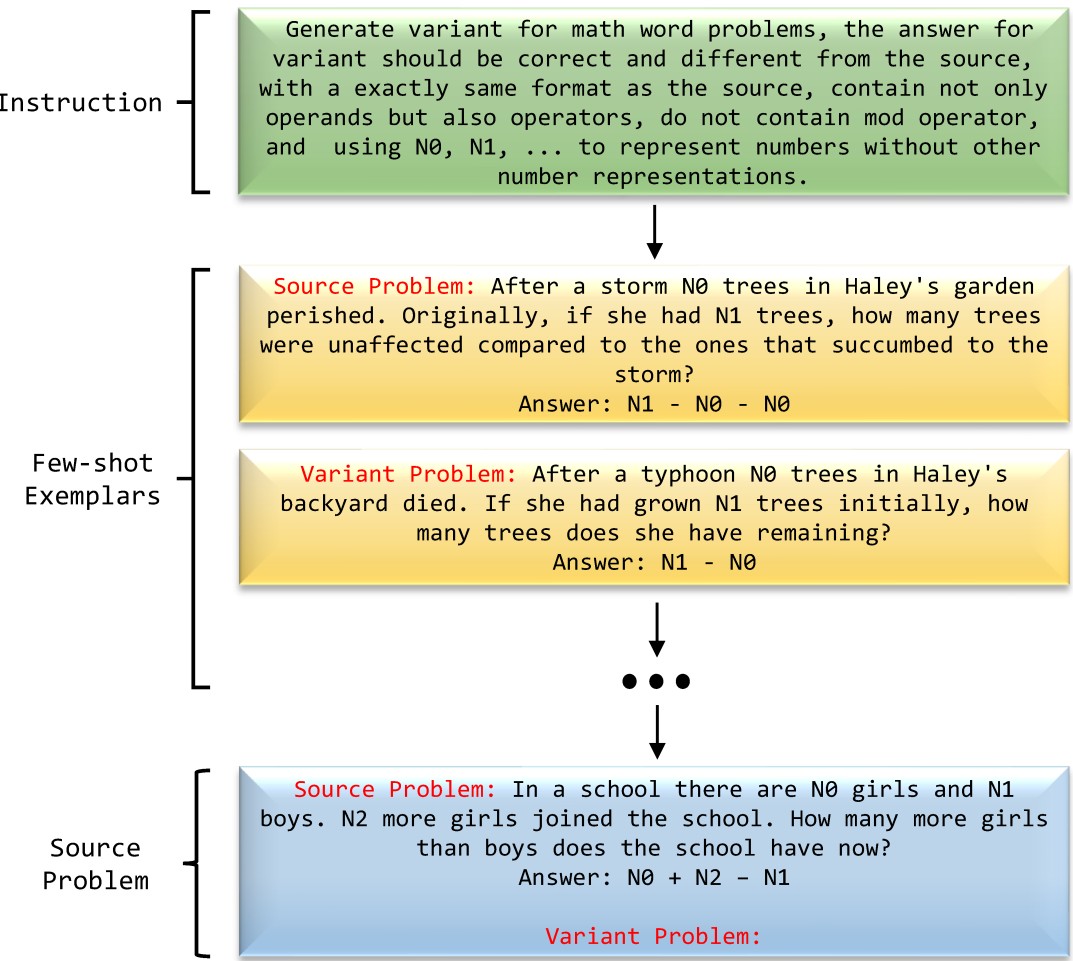

Figure 5: The few-shot prompts to generate problem-analogy MWP pairs.

| | Problem | Answer |
|---|---|---|
| Example 1 | Source Problem: After a storm N0 trees in Haley's garden perished. Originally, if she had N1 trees, how many trees were unaffected compared to the ones that succumbed to the storm? | N1 - N0 - N0 |
| | Variant Problem: After a typhoon N0 trees in Haley's backyard died. If she had grown N1 trees initially, how many trees does she have remaining? | N1 - N0 |
| Example 2 | Source Problem: An industrial machine made N0 shirts yesterday and N1 shirts today. It can make N2 shirts a minute. How many minutes did the machine work yesterday? | N0 / N2 |
| | Variant Problem: An industrial machine made N0 shirts yesterday and N1 shirts today. It can make N2 shirts a minute. How many minutes did the machine work in all? | (N0 + N1) / N2 |
| Example 3 | Source Problem: N0 red peaches, N1 yellow peaches, and N2 green peaches are in the basket. How many peaches are in the basket? | N0 + N1 + N2 |
| | Variant Problem: N0 red peaches, N1 yellow peaches, and N2 green peaches are in the basket. How many more red peaches than yellow peaches are in the basket? | N0 - N1 |

Table 5: The Full List of Few-Shot Examples