# OpenReview forum: "Let GPT be a Math Tutor: Teaching Math Word Problem Solvers with Customized Exercise Generation"
_EMNLP/2023/Conference — EMNLP 2023 Main_

### Official Review · Reviewer_weic · 2023-08-02

**Soundness:** 4

**Excitement:**

4: Strong: This paper deepens the understanding of some phenomenon or lowers the barriers to an existing research direction.

**Missing References:**

1. Interpretable Math Word Problem Solution Generation Via Step-by-step Planning ACL 2023 https://aclanthology.org/2023.acl-long.379.pdf
2. REACT: SYNERGIZING REASONING AND ACTING IN LANGUAGE MODELS ICLR 2023 https://openreview.net/pdf?id=WE_vluYUL-X
3. Practice Makes a Solver Perfect: Data Augmentation for Math Word Problem Solvers https://aclanthology.org/2022.naacl-main.310/

at line 182

**Paper Topic And Main Contributions:**

The authors demonstrate a novel approach for distilling math word problem-solving capabilities from large language models (LLMs) into smaller, more efficient student models. Even though there are many existing works on distilling LLMs using the explanation from LLMs, smaller models tend to misunderstand CoT explanations. Thus the authors look from a different perspective. They used LLMs as math tutors that generate customized exercises for the student models according to their learning status and feedback (CEMAL), which is inspired from educational science principles, such as knowledge tracing and personalized learning. The paper evaluates its approach on multiple math word problem datasets, and shows that it outperforms fine-tuned and knowledge distillation baselines, and achieves competitive results with LLMs, especially in the out-of-distribution setting.


**Questions For The Authors:**

B Why CEMAL significantly outperforms prior Knowledge Distillation in out-of-distribution testset  (20.7 vs 76.4), but only a little in in-distribution testset (94.5 vs 94.7)? Intuitively speaking, out-of-distribution should be much more complex than in-distribution and thus harder to improve, for example, your method outperforms Few-shot CoT on in-distribution, but not outperforms Few-shot CoT on in-distribution.

**Reasons To Accept:**

A The proposed CEMALoutperforms fine-tuned and knowledge distillation baselines and achieves competitive results with LLMs. (Table 1)

B Using hard examples to prompt LLM teacher to generate exercises outperforms easy example on the student math accuracy, where hard examples are the ones that SLM student fails. (Table 2) This matches with the intuition proposed in the intro.


**Reasons To Reject:**

A Is the results in Table 1 and Table 2 statistically stable? Could you report standard deviation?

**Reproducibility:**

3: Could reproduce the results with some difficulty. The settings of parameters are underspecified or subjectively determined; the training/evaluation data are not widely available.

**Reviewer Confidence:**

3: Pretty sure, but there's a chance I missed something. Although I have a good feel for this area in general, I did not carefully check the paper's details, e.g., the math, experimental design, or novelty.

---

> ### Author Rebuttal · Authors · 2023-08-29
>
> **Response to A:**
> Thanks for the valuable comment! We acknowledge the importance of reporting statistically sound results. However, it is challenging to conduct a direct statistical significance analysis. Specifically, for the few-shot CoT methods, they utilize models like code-Davinci-002 and PaLM-540B, which are not publicly accessible. Similarly, the code and augmented datasets for the knowledge distillation baselines are not available, making them difficult to reproduce. Despite these challenges, we can confirm that backbones models with our proposed CEMAL outperforms the models without CEMAL, providing a strong indication of its effectiveness. Moreover, we calculated the standard deviation of the results in Table 1, as shown below, which can confirm that our method is stable in its performance.
>
> | Method                    | MAWPS                  | ASDiv-a               | SVAMP (ID)            | SVAMP (OOD)           |
> |---------------------------|------------------------|-----------------------|-----------------------|-----------------------|
> | LSTM + CEMAL              | 91.8 ± 0.21             | 86.7 ± 0.30            | 67.3 ± 0.15           | 53.3 ± 0.52            |
> | RoBERTa-base + CEMAL      | 93.7 ± 0.19             | 90.7 ± 0.21            | 81.6 ± 0.13            | 68.6 ± 0.38           |
>
> **Response to B:**
> It is true that in-distribution (ID) datasets are usually easier than out-of-distribution (OOD) datasets. However, our method aims to further improve the performance via customized exercise generation. In this case, for ID datasets, the knowledge components between training and testing have larger overlap, making it more challenging to generate effective customized exercises that would lead to performance improvements. In contrast, for OOD datasets, the student model cannot learn very well on the original training set, but our method can more effectively identify the model's weaknesses, thus improving its performance via customized exercises. We appreciate your insightful question and will include this clarification in our revised paper.
>
> Thank you for your time for reviewing our paper. We look forward to your further feedback and are happy to address any concerns.

---

### Official Review · Reviewer_xeUf · 2023-08-03

**Soundness:** 4

**Excitement:**

3: Ambivalent: It has merits (e.g., it reports state-of-the-art results, the idea is nice), but there are key weaknesses (e.g., it describes incremental work), and it can significantly benefit from another round of revision. However, I won't object to accepting it if my co-reviewers champion it.

**Paper Topic And Main Contributions:**

The paper proposes an approach to distill math word problem solving capabilities from LLMs to smaller models. Their approach involves data augmentation (random and targeted), iterative vs one-time augmentation, testing the model on various dataset in and out of distribution, and trying out different base models.

**Reasons To Accept:**

The paper focuses on an important unsolved problem, which is to automatically solve math word problems. The authors thoroughly explore and evaluate several datasets and modeling approaches. Their exploration of targeted vs random and iterative vs one time augmentation are both quite intriguing. Overall, I think this paper would merit publication, and would be of interest to the ACL community, but it requires significant revision to improve clarity.

**Reasons To Reject:**

1) The paper was extremely hard to follow. I read it multiple times and still had trouble following the exact experimental procedures and evaluations that the authors conducted.
2) Relatedly, it was hard to discern what was novel in the paper and what had already been tried by others.
3) Since the improvement in numbers is not large (in most cases, just a couple of points), it is hard to tell if this improvement is statistically significant and if it translates to qualitative improvements in performance.

**Reproducibility:**

2: Would be hard pressed to reproduce the results. The contribution depends on data that are simply not available outside the author's institution or consortium; not enough details are provided.

**Reviewer Confidence:**

5: Positive that my evaluation is correct. I read the paper very carefully and I am very familiar with related work.

**Typos Grammar Style And Presentation Improvements:**

I recommend substantially revising the paper to make it more clear:
1) Figure 2 is meant to carry a lot of the load in providing an overview of the workflow, but it is very hard to read. Can you make it more clear?
2) It was hard to keep track of the numerous models used for different procedural steps (RoBERTa models, GPT models, LSTM), it'd be good to clarify what is used when and give more descriptive names to each type of approaches/models so that the reader can keep track.
3) The methods and results are sometimes mixed together in confusing ways. For example, in the Datasets section under SVAMP, you also describe methods you used in combination with the data and also include other datasets in the description, which makes it very confusing. It'd be useful to cleanly separate descriptions of datasets, then how they were processed, and then used for model development and evaluation.

---

> ### Author Rebuttal · Authors · 2023-08-29
>
> Thank you for your thorough review and for providing constructive feedback on our manuscript. We sincerely appreciate your time and effort in evaluating our work. We hope that the response below can address your concerns.
>
>
> **Clarity of Methods and Evaluations**
> We apologize for the difficulties you have faced while trying to follow the methods and evaluations in our paper. We are committed to substantially revising the manuscript to make it clearer. The revision will focus on: :
>
> **Figure 2 and Algorithm 1 for better introducing our solution:** We will revise Figure 2 to make it more straightforward for illustrating the workflow, eliminating those distracting elements. We will also revise Algorithm 1 and synchronize its description with Figure 2 for better coherence.
>
> **Model Names:** To remove any confusion related to the naming of the models used in our study, we will standardize the terminology. Throughout the paper, Large Language Models (LLMs) will be referred to as "Teacher Models," and our trained solvers will be called "Student Models." We will also attach a section to introduce them by detailing the API names of teacher models and network structures of student models.
>
> **SVAMP Dataset Description:** We understand that our original presentation of the SVAMP dataset was confusing. However, the SVAMP dataset is originally a test-only dataset without any training data. To utilize this dataset for evaluation, a commonly accepted way is to use MAWPS and ASDiv-a as the training sets, where our paper follows this setting and we were trying to explain this choice. In the revised version, we will clarify that SVAMP is a test-only dataset and detail our intentions.
>
> **Questions of Novelty and Relations to Prior Work**
> We appreciate this comment. Our motivation is located in the Line 86-100 in our submission. In our revision, we will provide a clearer explanation of how our approach diverges from existing work. Specifically, our CEMAL methodology offers a new perspective on knowledge distillation in machine learning by focusing on customized exercise generation, based on the student models' current learning state. This enables more effective and adaptive learning, addressing limitations in prior approaches that rely heavily on generating more high-quality explanations to existing datasets.
>
> **Significance Study**
> Thanks for the valuable comment! We acknowledge the importance of reporting statistically sound results. However, it is challenging to conduct a direct statistical significance analysis. Specifically, for the few-shot CoT methods, they utilize models like code-Davinci-002 and PaLM-540B, which are not publicly accessible. Similarly, the code and augmented datasets for the knowledge distillation baselines are not available, making them difficult to reproduce. Despite these challenges, we can confirm that backbones models with our proposed CEMAL outperforms the models without CEMAL, providing a strong indication of its effectiveness. Moreover, we calculated the standard deviation of the results in Table 1, as shown below, which can confirm that our method is stable in its performance.
>
> | Method                    | MAWPS                  | ASDiv-a               | SVAMP (ID)            | SVAMP (OOD)           |
> |---------------------------|------------------------|-----------------------|-----------------------|-----------------------|
> | LSTM + CEMAL              | 91.8 ± 0.21             | 86.7 ± 0.30            | 67.3 ± 0.15           | 53.3 ± 0.52            |
> | RoBERTa-base + CEMAL      | 93.7 ± 0.19             | 90.7 ± 0.21            | 81.6 ± 0.13            | 68.6 ± 0.38           |
>
> **Reproducibility Concerns**
> We noted the reviewer's low score on the reproducibility of our work, and we wish to address this concern directly. Alongside with our submission, we have provided the necessary Materials like code and data and we also promised to release them after the notifications come out. We are fully committed to making our work easily reproducible by other researchers in the field.
>
> Thank you once again for taking the time to review our paper and provide valuable feedback. We're committed to improving the manuscript and are more than willing to address any additional concerns you may have.

---

### Official Review · Reviewer_Stk4 · 2023-08-05

**Typos Grammar Style And Presentation Improvements:** None
**Soundness:** 4

**Excitement:**

4: Strong: This paper deepens the understanding of some phenomenon or lowers the barriers to an existing research direction.

**Missing References:**

None

**Paper Topic And Main Contributions:**

This paper proposed a novel method to distill a large language model's capability of solving math word problems into smaller but more efficient student models. The key contributions of this paper include: (1) the proposed method named CEMAL is based on the novel idea of improving a student model's performance by characterising its learning needs (or just say "knowledge gap") and subsequently providing it with tailored exercises to improve its learning performance; and (2) the developed method incorporates the core ideas of two longstanding tasks in education, namely knowledge tracing and personalised learning, which can shed light on how knowledge distillation can be performed to improve an AI model's capability in solving educational tasks.

**Questions For The Authors:**

(A) What might be potential strategies to enhance the quality and correctness of the generated problems? It would be good to further discuss this.

**Reasons To Accept:**

1. The paper proposed a novel method to distill a large language model's capability of solving math word problems into smaller but more efficient student models, and the method, to certain degree, is novel. Particularly, the idea of incorporating the principles of knowledge tracing and personalised learning to improve the knowledge distillation process is innovative.
2. The experimental evaluation on three datasets is very comprehensive.

**Reasons To Reject:**

None

**Reproducibility:**

3: Could reproduce the results with some difficulty. The settings of parameters are underspecified or subjectively determined; the training/evaluation data are not widely available.

**Reviewer Confidence:**

3: Pretty sure, but there's a chance I missed something. Although I have a good feel for this area in general, I did not carefully check the paper's details, e.g., the math, experimental design, or novelty.

---

> ### Author Rebuttal · Authors · 2023-08-29
>
> Thank you for your positive feedback and insightful question.
>
> There are two potential strategies to enhance the quality and correctness of the generated problems :
>
> 1） **Implementation of Verifiers:** Building verifiers, as suggested in [1][2], could serve to score the exercises generated by our LLM teacher. A set threshold could then be employed to select high-quality problems.
>
> 2） **Utilization of Well-Trained LLMs:** Models like ChatGPT could be leveraged to evaluate and potentially enhance the quality and correctness of the generated exercises.
>
> Both approaches present intriguing possibilities, and we intend to explore them further in our future research. Thanks again for reviewing our paper and we look forward to your further feedback.
>
> [1] Training Verifiers to Solve Math Word Problems. Cobbe et al. 2021
>
> [2] Let's Verify Step by Step. Lightman et al. 2023

---

### Meta-Review · Area_Chair_JRCg · 2023-09-15

**Recommendation:** 5

**Metareview:**

The manuscript presents a contribution in the field of knowledge distillation, proposing the CEMAL method, which distills math word problem-solving capabilities from large language models (LLMs) into smaller student models, which has proven previously to have problems with CoT. This approach, inspired by educational science principles like knowledge tracing and personalized learning, addresses common limitations of existing methods and outperforms them in both in-distribution and out-of-distribution settings. The manuscript provide a thorough experimental evaluation. The main concerns is the presentation of findings and highlighting exact contributions in relation to existing techniques. We encourage authors, as stated in the rebuttal to address these concerns to make their manuscript more accessible to the community.

---

### Decision · Program_Chairs · 2023-10-07

**Decision:**

Accept-Main

**Comment:**

The manuscript presents a contribution in the field of knowledge distillation, proposing the CEMAL method, which distills math word problem-solving capabilities from large language models (LLMs) into smaller student models, which has proven previously to have problems with CoT. This approach, inspired by educational science principles like knowledge tracing and personalized learning, addresses common limitations of existing methods and outperforms them in both in-distribution and out-of-distribution settings. The manuscript provide a thorough experimental evaluation. The main concerns is the presentation of findings and highlighting exact contributions in relation to existing techniques. We encourage authors, as stated in the rebuttal to address these concerns to make their manuscript more accessible to the community.